# Novel Phosphorus-Nitrogen-Containing Ionic Liquid Modified Metal-Organic Framework as an Effective Flame Retardant for Epoxy Resin

**DOI:** 10.3390/polym12010108

**Published:** 2020-01-05

**Authors:** Rong Huang, Xiuyan Guo, Shiyue Ma, Jixing Xie, Jianzhong Xu, Jing Ma

**Affiliations:** The Flame Retardant Material and Processing Technology Engineering Research Center of Hebei Province, Hebei University, Baoding 071002, China; huangrong_hbu@163.com (R.X.); guoxiuyan_hbu@163.com (X.G.); mashiyue123@163.com (S.M.); Xiejx_hbu@163.com (J.X.)

**Keywords:** MOF, ionic liquid, epoxy resin, flame retardant

## Abstract

Metal-organic frameworks (MOFs) have shown great potential in flame retardant applications; however, strategies for fully exploiting the advantages of MOFs in order to further enhance the flame retardant performance are still in high demand. Herein, a novel MOF composite was designed through the generated cooperative role of MOF (NH_2_-MIL-101(Al)) and a phosphorus-nitrogen-containing ionic liquid ([DPP-NC_3_bim][PMO]). The ionic liquid (IL) was composed of imidazole cation modified with diphenylphosphinic group (DPP) and phosphomolybdic acid (PMoA) anions, which can trap the degrading polymer radicals and reduce the smoke emission. The MOF acts as a porous host and can avoid the agglomeration of ionic liquid. Meanwhile, the -NH_2_ groups of NH_2_-MIL-101(Al) can increase the compatibility with epoxy resin (EP). The framework is expected to act as an efficient insulating barrier to suppress the flame spread. It was demonstrated that the MOF composite (IL@NH_2_-MIL-101(Al)) is able to effectively improve the fire safety of EP at low additions (3 wt. %). The LOI value of EP/IL@NH_2_-MIL-101(Al) increased to 29.8%. The cone calorimeter results showed a decreased heat release rate (51.2%), smoke production rate (37.8%), and CO release rate (44.8%) of EP/IL@NH_2_-MIL-101(Al) with respect to those of neat EP. This strategy can be extended to design other advanced materials for flame retardant.

## 1. Introduction

Epoxy resin (EP) has proven to be one of the most important thermosetting resins due to its low cost, convenient curing processes, electrical insulation and excellent chemical resistance [1,2]. Currently, EP is widely used in adhesives, composites, coatings and casting materials [3]. Unfortunately, EP exhibits high flammability, and will produce lots of poisonous gases in the burning process, which is a great threat to human safety [4,5,6]. A potential strategy for enhancing the fire resistance and suppressing the smoke emission of EP is to develop efficient flame retardants, which can be added to EP in order to enhance its performance [7,8,9]. Therefore, the development of effective methods for preparing flame retardants is of great importance in improving the fire resistance of EP.

Metal-organic frameworks (MOFs) are a typical kind of nanoporous crystalline structure comprising a metal cluster and an organic ligand [10]. They have offered great advantages in various applications due to their highly tunable chemical properties and pore structrues [11,12]. Recently, several MOFs have been investigated as flame retardants in polymers [13,14,15,16,17,18,19]. For instance, Hu et al. [15] synthesized iron-based and cobalt-based MOFs and investigated the flame-retardant properties of MOF/polystyrene composites. The results indicated that the improved flame-retardant performance of MOF/polystyrene composites was probably due to the thermal insulation and char formation effects of MOF. Meanwhile, the flame retardant performance of a single MOF is often limited. To fully exploit the advantages of MOFs in flame retardant application, the design of MOF composites by combining MOF with other functional materials is an effective method [20]. Thus, the MOF composite could combine the advantages of the individual components and exhibit new properties that the individual parts did not possess. For instance, Xu et al. [18] synthesized a ZIF-8/RGO composite. The results indicate that ZIF-8/RGO effectively improved the fire resistance of EP by means of the synergistic effect of ZIF-8 and RGO. Meanwhile, combinations of other materials with MOF in order to produce MOF composites for flame retardant application are limited and in high demand.

Ionic liquids (ILs), comprising a cation and an anion ion, are widely used in various fields due to their high ionic conductivity, wide electric potential range, negligible volatility and low flammability [21,22]. Several works have indicated that ILs exhibit unique properties in flame retardant fields [23,24,25,26,27,28,29]. For instance, Yang et al. [23] synthesized a phosphorus-containing IL. The IL showed a synergistic effect with ammonium polyphosphate (PP), and the limiting oxygen index (LOI) of PP/IFR (30 wt. %) reached 31.8%. Li et al. [24] prepared a polyoxometalate-based ionic liquid. The IL exerted a positive effect in increasing the flame retardant properties of PP. Shi et al. [25] designed a phosphorus-containing ionic liquid ([Dmim]Tos), and the addition of 4 wt. % [Dmim]Tos in EP effectively increased the LOI value to 32.5%. In these studies, the ILs were added directly to the polymers. However, the viscosity of IL makes it difficult to disperse, thus neutralizing the effect of IL [20]. Meanwhile, the properties of a single IL are limited. Therefore, strategies for fully exploring the inherent potential and enhancing the performance of IL as flame retardant are in urgent demand.

In this work, a novel IL-modified MOF composite was designed combining the advantages of IL and MOF, while at the same time exhibiting excellent flame retardant properties (Scheme 1). To demonstrate this strategy, NH_2_-MIL-101(Al) was prepared as the MOF material. On one hand, the metal cluster of NH_2_-MIL-101(Al) decreases the smoke emission and catalyze the char-forming process. On the other hand, –NH_2_ in the organic ligand of NH_2_-MIL-101(Al) is helpful for increasing compatibility with EP, and provides flame-retardant elements such as nitrogen and aluminum. For one thing, the –NH_2_ group is able to react with the epoxy group, inducing anionic homopolymerization in epoxy resin [1]. For another thing, due to the presence of –NH_2_, NH_3_ will be released during the combustion of the epoxy resin, thus diluting the combustible gas and playing a flame-retardant role in the gas phase. The aluminum element is able to produce compact coke during the burning process of the epoxy resin, which plays a very good role in protecting and suppressing the smoke [2]. For the IL, we designed a phosphorus-nitrogen-containing IL composed of imidazole cations modified with the diphenylphosphinic group (DPP) and phosphomolybdic acid (PMoA) anions. As is well known, N-methyl imidazole increases compatibility with EP [25]. Meanwhile, DPP and phosphomolybdic acid exhibits good flame retardant properties [29,30]. Based on the unique properties of NH_2_-MIL-101(Al) and IL, the IL-modified NH_2_-MIL-101(Al) composite (IL@NH_2_-MIL-101(Al)) could combine the advantages of the two parts. The flame-retardant effect of IL@NH_2_-MIL-101(Al) on EP is measured, and a possible mechanism is also proposed.

## 2. Experimental

### 2.1. Materials

Aluminum chloride hexahydrates (AlCl_3_•6H_2_O, 99%) was obtained from Guangfu Fine Chemical Research Ins. (Tianjin, China). 1-methylimidazole, 2-aminoterephthalic acid, 3-bromopropylaminehydrobromide (98%), potassium hydroxide (KOH, 85%), trimethylamine (TEA), and diphenylphosphinic chloride (DPP-Cl) were obtained from J&K Scientific Co., Ltd. (Beijing, China). N,N-dimethyformamide (DMF), anhydrous ethanol, phosphomolybdic acid, tetracosahydrate, and methanol were purchased from Comiao Chemical Reagent Co., Ltd. (Tianjin, China). Ethylacetate was purchased from Beichen Fangzheng Century Co., Ltd. (Tianjin, China). m-phenylenediamine was purchased from Aladdin Industrial Corporation (Shanghai, China). EP (DGEBA, E-44) was kindly provided by Baling Petrochemical Corporation Branch, Assets Management Corporation, China Petrochemical Corporation (Yueyang, Hunan, China). 

### 2.2. Synthesis of [NH_2_C_3_bim][PMo]

The synthesis reaction is shown in Scheme 2. [NH_2_C_3_bim][PMo] was synthesized by an ion exchange reaction using [NH_2_C_3_bim][Br] and PMoA. [NH_2_C_3_bim][Br] was prepared by a reaction of 1-methylimidazole (7.9 mL) with 3-bromopropylamine hydrobromide (21.892 g) in anhydrous ethanol (250 mL) under nitrogen atmosphere [20]. After stirring for 24 h at 303 K, ethanol was removed using the rotary evaporation method. The product was further dissolved in water and the pH was adjusted to 8 by the addition of KOH. The produced KBr precipitate was removed by centrifugation. Then [NH_2_C_3_bim][Br] was obtained by extraction method using ethyl acetate. ^1^H NMR (400 MHz, DMSO-d_6_) δ 9.35 (s, 1H ), 8.11 (s, 2H), 7.91 (s, 1H), 7.83 (s, 1H), 4.39 (t, *J* = 8.0 Hz, 2H), 3.91 (s, 3H), 2.86 (s, 2H), 2.18 (t, *J* = 4.0 Hz, 2H); ^13^C NMR (100 MHz, DMSO-d_6_): δ 136.8, 123.8, 122.2, 46.0, 36.0, 35.7, 27.6.

Finally, the [NH_2_C_3_bim][PMo] was synthesized by an ion exchange reaction using [NH_2_C_3_bim][Br] and PMoA of an equal molar ratio in ethanol for 10 h with stirring.

### 2.3. Synthesis of [DPP-NC_3_bim][PMo]

[NH_2_C_3_bim][PMo] (1.523 g), DMF (300 mL), and 12 drops of TEA (catalyst) were added in a 500 mL three-neck flask. DPP-Cl (0.387 mL) was dispersed in DMF (60 mL) and then slowly added to the above flask [30]. The reaction solution was reacted in an ice-water bath for 1 h under N_2_ atmosphere with stirring. Then the solution was further reacted at 353 K for 24 h. The obtained product was designated as [DPP-NC_3_bim][PMo].

### 2.4. Synthesis of NH_2_-MIL-101(Al)

NH_2_-MIL-101(Al) was prepared based on a previous study [31]. Briefly, AlCl_3_·6H_2_O (1.02 g) and 2-aminoterephthalic acid (1.12 g) were dispersed in 60 mL DMF. Then the solution was placed in a 100 mL autoclave and reacted at 130 °C for 72 h. When the reaction temperature cooled to room temperature, the yellow precipitate was recovered by centrifugation, and further washed several times with ethanol. Finally, the obtained NH_2_-MIL-101(Al) was dried at 80 °C in a vacuum oven.

### 2.5. Synthesis of [DPP-NC_3_bim][PMo]@NH_2_-MIL-101(Al)

The composite material was obtained by incorporating [DPP-NC_3_bim][PMo] (1.6 g) into the reaction solution of NH_2_-MIL-101(Al). Other procedures were similar to the synthesis method of NH_2_-MIL-101(Al). The obtained product was simplified as IL@NH_2_-MIL-101(Al).

### 2.6. Synthesis of EP Composites

First, EP (80 g) was heated to 60 °C while stirring, and the bubbles were removed by vacuum and ultrasonic treatment. Then 3 wt. % of the prepared flame retardants (NH_2_-MIL-101(Al) and IL@NH_2_-MIL-101(Al)) were added into the EP to analyze flame retardant performance. Herein, M-phenylenediamine (8.8 g), acting as the curing agent, was mixed together with the EP solution. Then the EP composites were poured rapidly into a mold and the mold was held in a vacuum oven at 60 °C for 20 min. The EP composites were thermally cured at 80 °C for 2 h and then 150 °C for 3 h. The neat EP sample was prepared using the same method without the addition of flame retardant. The obtained materials were written as EP, EP/3.0NH_2_-MIL-101(Al) and EP/3.0IL@NH_2_-MIL-101(Al), in which the number represents the weight percentage of flame retardant in the EP.

### 2.7. Characterizations

Powder X-ray diffractometer (XRD) analyses were performed using D8ADVANCE (Bruck, Karlsruhe, Germany). The scanning rate was 5°·min^−1^ in the range of 3~50°. Fourier transform infrared (FTIR) spectra were measured on a Tensor-27 (Bruck, Karlsruhe, Germany) using KBr pellets. Field emission scanning electron microscopy (SEM) was carried out on a JSM-7500F (JEOL, Japan). Transmission electron microscopy (TEM) analysis was used to assess the size and morphology of the particles (Tungsten filament, JEM1200EX, JEOL, Japan). N_2_ adsorption was carried out at 77 K with a Micromeritics BET surface area analyzer to determine the surface area of the particles (Tristar||3020, Micromeritics Instrument Ltd. Shanghai, China). X-ray photoelectron spectroscopy (XPS) was carried out on a Thermo Fisher K-alpha (Massachusetts, USA). ^1^H and ^13^C NMR spectra were gained on a BUXI-I NMR −400 MHz spectrometer (Oxford spectrum, Shanghai, China), and the solvent was DMSO-d_6_. Thermogravimetric analysis (TGA) was performed on a STA449CQMS403C (NETZSCH, Selb, Germany). The samples were heated from room temperature to 800 °C at a heating rate of 10 °C·min^−1^ under N_2_ atmosphere. The LOI of the samples were tested on a HC-3 oxygen index meter (Jiangning Analysis Instrument Company, Jiangning, China). The dimensions of the tested sample were 130 mm × 6 mm × 3 mm. The combusion behaviors of the samples were analyzed by cone calorimeter using FTT CONE (Fire Testing Technology, East Grinstead, London, UK) equipment. The samples with a size of 100 mm × 100 mm × 3 mm were tested at a heat flux of 50 kW·m^−2^.

## 3. Results and Discussion

### 3.1. Characterization of the Materials

The crystal structures were examined by XRD measurement. Figure 1 shows the XRD patterns of NH_2_-MIL-101(Al) and IL@NH_2_-MIL-101(Al). All the peaks are sharp and clear in the XRD curves. The pattern of NH_2_-MIL-101(Al) is well consistent with the reported peaks, indicating that the crystal structure was successfully obtained [31,32]. The almost identical peaks of IL@NH_2_-MIL-101(Al) and NH_2_-MIL-101(Al) demonstrate that the crystal structure of NH_2_-MIL-101(Al) in the composite material did not change during the in situ crystallization.

FT-IR was conducted to further analyze the chemical bonds of the obtained materials. The results are shown in Figure 2. For NH_2_-MIL-101(Al), the characteristic peaks at 3495 cm^−1^, 3385 cm^−1^, 1686 cm^−1^, 1600 cm^−1^ and 1336 cm^−1^ show the asymmetric and symmetric stretching of amine groups in the ligand, and the N–H and typical C–N stretching in the aromatic amines, respectively [31]. Additionally, two new characteristic adsorption peaks of IL@NH_2_-MIL-101(Al) appear at 950 cm^−1^ and 1259 cm^−1^, corresponding to P–N stretching vibration and P=O vibration, respectively [30]. These absorption peaks demonstrate the presence of the functional groups in the phosphorus-nitrogen-containing ionic liquid. Additionally, the characteristic peaks of NH_2_-MIL-101(Al) do not change much in the spectrum of IL@NH_2_-MIL-101(Al), indicating the structural integrity of NH_2_-MIL-101(Al) throughout the reaction processes. The above results demonstrate the successful synthesis of NH_2_-MIL-101(Al) and IL@NH_2_-MIL-101(Al).

To further illuminate the structure of [DPP-NC_3_bim][PMo], ^1^H and ^13^C NMR (Figure 3) were used to distinguish the H and C chemical environments in [NH_2_C_3_bim][Br]. As shown with ^13^C NMR, the carbon resonance peaks coming from the imidazole ring appear at 136.8, 123.8, and 122.2 ppm. The carbon signal of 46.0 ppm can be ascribed to the methyl of N-methylimidazole. The carbon resonance peaks located at 36.0, 35.7, 27.6 ppm can be attributed to the propyl group. Furthermore, all of the characteristic hydrogen resonance peaks of [NH_2_C_3_bim][Br] can be observed in its ^1^H NMR spectrum. The chemical shift at 3.91 ppm (3H) is related to methyl in N-methylimidazole. The signals at 9.35, 7.91, and 7.83 ppm can be attributed to the imidazole ring. Then the signals derived from the propyl amine group correspond to the shifts at 8.11, 4.39, 2.86, and 2.18 ppm. These results directly indicate that the synthesis of [NH_2_C_3_bim][Br] was successful. Next, the XPS spectra were carried out, and the corresponding N 1s high-resolution spectra of IL is shown in Figure 4. In the N 1s high-resolution spectra of IL, the binding energies at 401.55 and 398.68 eV are due to P-NH-CH_2_ and the tertiary amine group, respectively. Therefore, the structure of IL is confirmed. In addition, the peaks of the C 1s, N 1s, and O 1s elements all appear in the XPS spectra of IL, NH_2_-MIL-101(Al), and IL@NH_2_-MIL-101(Al). For IL@NH_2_-MIL-101(Al), the signals of the P 2p and Mo 3d elements derived from the IL and Al 2p elements originated from NH_2_-MIL-101(Al) and also appear in the XPS spectrum, indirectly confirming the successful modification of IL into NH_2_-MIL-101(Al), wherein the content of P is 1.02%. The corresponding IL content in NH_2_-MIL-101(Al) is 23.2%.

The morphology of the NH_2_-MIL-101(Al) and IL@NH_2_-MIL-101(Al) composites were investigated by SEM and TEM. As shown in Figure 5, NH_2_-MIL-101(Al) presents as an aggregated polyhedral. The size is about 150 nm, and the surface is smooth. For the IL@NH_2_-MIL-101(Al) composite, the surface is rough, but the shape is similar to that of the NH_2_-MIL-101(Al), indicating that the crystal morphology of NH_2_-MIL-101(Al) was not affected by the introduction of ionic liquid.

Figure 6 displays the N_2_ adsorption isotherm of NH_2_-MIL-101(Al) and IL@NH_2_-MIL-101(Al). This isotherm illustrates the typical steps of the MIL-101 structure. First, at very low relative pressures (P/P_0_ < 0.05), only the supertetrahedra are filled. Then, with the increase in pressure, the medium (P/P_0_ = 0.15) are filled, and later, the large cavities (P/P_0_ = 0.20) [31]. For NH_2_-MIL-101(Al), the specific BET surface area is 1870 m^2^/g. However, the specific BET surface area of IL@NH_2_-MIL-101(Al) is reduced to 422 m^2^/g, which is due to the fact that the ionic liquid was encapsulated into the pore channel of NH_2_-MIL-101(Al), indicating the successful modification of NH_2_-MIL-101 (Al) by IL.

### 3.2. Morphological Analysis of EP and EP Composites

The cross sections of EP and EP composites were characterized by SEM in order to obtain the dispersion characteristics of the added flame retardants in EP. The results are shown in Figure 7. The cross sections of EP composites are rougher than the cross sections of EP due to the added flame retardants. For the EP/NH_2_-MIL-101(Al) composite, most of the NH_2_-MIL-101(Al) particles are well dispersed in the epoxy matrix. A small part of agglomerated NH_2_-MIL-101(Al) nanoparticles is present in Figure 7b. For the EP/IL@NH_2_-MIL-101(Al) composites, the IL@NH_2_-MIL-101(Al) particles are uniformly and randomly dispersed within the EP matrix (shown in Figure 7c). The good compatibility between IL@NH_2_-MIL-101(Al) and EP may be attributed to two reasons. On the one hand, the amino functional groups of NH_2_-MIL-101(Al) increase affinity with EP [33]. On the other hand, the N-methyl imidazole part of IL are able to promote the curing processes of EP, leading to enhanced compatibility with EP [25]. The above results indicate that the combination of IL and NH_2_-MIL-101(Al) is beneficial for good compatibility with EP.

EDS analysis was carried out to explain the results of the elemental analysis of the EP/ NH_2_-MIL-101(Al), and EP/IL@ NH_2_-MIL-101(Al) materials. The results of the EDS analysis are shown in Table 1. Based on the results of the analysis, four main elements of C, O, N and Al are present in EP/ NH_2_-MIL-101(Al). Due to the addition of ionic liquid, two new elements (P and Mo) are found in EP/IL@ NH_2_-MIL-101(Al).

### 3.3. Thermal Stability of EP and EP Composites

The thermal stability of the samples was investigated using the TGA method in a nitrogen atmosphere. The results of the samples are shown in Figure 8. The decomposition data obtained from Figure 8, containing the temperature at 5.0% weight loss (*T*_5%_), the temperature at the maximum decomposition rate (*T*_max_), and the residual mass at 800 °C (R), are shown in Table 2. Throughout the whole heating process, all of the samples have only a single decomposition stage under the N_2_ atmosphere. From the decomposition data shown in Table 2, the EP/NH_2_-MIL-101(Al) and EP/IL@NH_2_-MIL-101(Al) composites exhibit lower *T*_5%_ and *T*_max_ than those of EP, demonstrating that the decomposition process of EP can be promoted by the addition of NH_2_-MIL-101(Al) and IL@NH_2_-MIL-101(Al). Additionally, the EP/IL@NH_2_-MIL-101(Al) composite exhibits lower *T*_5%_ and *T*_max_ values (340.2 °C and 354.0 °C, respectively) than those of EP/NH_2_-MIL-101(Al) (351.6 °C and 371.5 °C, respectively). The above results indicate that combining NH_2_-MIL-101(Al) and IL can accelerate the decomposition process of EP. In terms of the residue, the EP/NH_2_-MIL-101(Al) and EP/IL@NH_2_-MIL-101(Al) composites show a greater mass of residue than that of EP, with increases of 18.0% and 76.4%, respectively, which can be attributed to the catalyzed char formation effect by IL and NH_2_-MIL-101(Al).

### 3.4. Flame Retardant Performance of EP Composites

LOI is a commonly used value for evaluating the flame retardant performance of polymers, and is the minimum oxygen concentration required for a material to catch flame under specific conditions [34]. In general, polymers with higher LOI will exhibit more effective flame retardant performances. Under the same conditions, the neat EP shows the lowest LOI value of 25.7%. The addition of 3.0 wt. % NH_2_-MIL-101(Al) to the EP matrix induces an increased LOI value (29.2%) and effectively decreases the falling phenomenon in the flames. The phosphorus-nitrogen-containing IL-modified NH_2_-MIL-101(Al) further increases the LOI value to as high as 29.8%, demonstrating the obviously enhanced flame retardant properties of EP.

In order to further analyze the combustion behavior of the EP and EP/IL@NH_2_-MIL-101(Al) composites, a cone calorimetry test was performed. The results are shown in Figure 9 and Table 3. Total heat release (THR) and heat release rate (HRR) are two parameters that can be used to evaluate the heat release process during combustion [35]. As shown in Figure 9a,c, the integration of NH_2_-MIL-101(Al) and IL@NH_2_-MIL-101(Al) in the EP matrix is able to reduce the values of THR and HRR. With an addition of NH_2_-MIL-101(Al) of 3.0 wt. %, the values of THR and HRR decrease to 110.79 MJ/m^2^ and 851.64 Kw/m^2^, respectively. Under the same conditions, the addition of IL@NH_2_-MIL-101(Al) results in lower values of THR and HRR (101.92 MJ/m^2^ and 585.72 Kw/m^2^, respectively), which are reduced by 14.39% and 51.24%, respectively, indicating the positive effect of IL in the combustion of EP. Meanwhile, IL and NH_2_-MIL-101(Al) have a synergistic effect on decreasing heat release.

Smoke emission during material combustion seriously endangers human life. It is another essential aspect in fire safety evaluation [36]. The total smoke release (TSP) and the smoke release rate (SPR) were evidently suppressed by the introduction of NH_2_-MIL-101(Al) and IL@NH_2_-MIL-101(Al). As shown in Figure 9b,d, for the EP/3.0IL@NH_2_-MIL-101(Al) composite, the total smoke release and the smoke release rate decreased by 13.07% and 37.84%, indicating the excellent smoke suppression performance of IL@NH_2_-MIL-101(Al). The individual NH_2_-MIL-101(Al) also performed well, with the total smoke release and the smoke release rate decreasing by 7.16% and 21.62% for the EP/NH_2_-MIL-101(Al) composite. The suppressed smoke release was possibly due to the catalytic role of the metal cluster and the barrier effect on EP. With respect to the excellent smoke suppressing effect of IL@NH_2_-MIL-101(Al), NH2-MIL-101(Al) induces a similar effect in the IL@NH_2_-MIL-101(Al) composite. Meanwhile, the IL is able to improve the gas phase function of IL@NH_2_-MIL-101(Al).

CO and organic carbon are the main emissions in fire disasters. Therefore, the release behavior of CO and organic carbon are two important parameters for evaluating material safety. Figure 9e,f shows the CO release rate and the total organic carbon yield results. The addition of the IL@NH_2_-MIL-101(Al) composite to EP is obviously able to reduce the CO release rate (44.83%) and the total organic carbon yield (14.38%) compared with those of EP. The suppressed emission phenomenon of CO and organic carbon could greatly reduce the threat to human safety represented by fire disasters. With respect to the performance of NH_2_-MIL-101(Al) on EP, the peak CO release rate and total organic carbon yield are 0.02 g/s and 73.86 g, respectively. These results also confirm the synergistic effect of IL and NH_2_-MIL-101(Al).

The flame-retardant effects of some other phosphorus-nitrogen-containing materials are summarized in Table 4. As can be seen from the comparison, we have synthesized a highly efficient flame retardant. When the addition of IL@NH_2_-MIL-101(Al) is 3 wt. %, it not only achieves a remarkable reduction in heat release and an obvious smoke suppression effect, but it also greatly reduces the release of toxic gas. To our knowledge, this is the first report of IL-modified MOF composite as an effective flame retardant. Therefore, the modification method provides a new way of preparing other high-efficiency flame retardants.

### 3.5. Analysis of Char Residue

SEM was applied to measure the microtopography of the char residues and to analyze the effect of the flame retardant on the char formation processes. Generally, the more char residues that are formed throughout the combustion process, the fewer the combustible materials, and the formed char can also work as a barrier for heat and smoke transfer [25]. As can be seen from Figure 10, the char residue of EP exhibits a discontinuous layer on its outside surface. Additionally, many little holes were generated on the internal surface. Following the addition of NH_2_-MIL-101(Al) to the EP matrix, the char layer on the exterior surface becomes more continuous, although holes still exist on the interior surface. For the EP/IL@NH_2_-MIL-101(Al) composite, the char layer becomes more compact than those of EP and EP/NH_2_-MIL-101(Al). A continuous char layer is able to resist the transfer of heat, smoke and combustible materials, thus suppressing the development of fire in EP. The above results indicate that the IL@NH_2_-MIL-101(Al) composite is able to promote the char-forming processes of EP, and this property was attributed to the combined effect of NH_2_-MIL-101(Al) and IL.

Raman spectroscopy is a widely used technique for measuring the degree of graphitization of char residues. Char residues with a higher degree of graphitization possess better thermostability properties [13,25]. As exhibited in Figure 11, the characteristic peaks at the wavenumbers of 1360 cm^−1^ and 1600 cm^−1^ represent the D and G bands, respectively. The D band represents the vibrations of the disordered carbon compounds, and the G band was generated from the graphite layers. A higher I_D_/I_G_ ratio denotes a lower degree of graphitization. As can be observed from Figure 11, the EP/IL@NH_2_-MIL-101(Al) composite shows the highest degree of graphitization of char residues, demonstrating a more effective char layer formed than those of EP and EP/NH_2_-MIL-101(Al). The improved char-forming effect of IL@NH_2_-MIL-101(Al) on EP is consistent with the flame-retardant results of EP and the EP composites.

### 3.6. Proposed Mechanism

The possible mechanisms by which IL@NH_2_-MIL-101(Al) could improve the fire resistance of EP are illustrated in Scheme 3. For the neat EP, large amounts of heat, smoke and CO are released in cases of fire, which is a great fire hazard for EP. The addition of IL@NH_2_-MIL-101(Al) greatly decreases the release of heat, smoke and CO of EP. Based on the above analysis of combustion performance and char residue product, the flame-retardant mechanism of IL@NH_2_-MIL-101(Al) on EP takes place in two phases. In the gas phase, the phosphorus-containing compounds of the diphenylphosphinic group in the imidazole cation of IL react with active free radicals in order to restrain the flame spread. Thus, the release of heat and CO can be effectively reduced. Meanwhile, the addition of NH_2_-MIL-101(Al) is also able to reduce the emission of heat and CO of EP due to the catalytic oxidation effect of CO by the metal cluster in NH_2_-MIL-101(Al). In the condense phase, the imidazole cation and phosphomolybdic acid anion of IL act as catalyst, inducing the cross-linking reaction in EP composites and trapping the degrading polymer radicals; thus, the formation of char can be promoted. Additionally, the framework of NH_2_-MIL-101(Al) combined with the char layer formed by IL are effective barriers to reducing the transfer of heat and the exposure of the EP matrix to the heat source. These two phase effects lead IL@NH_2_-MIL-101(Al) to have good fire-resistance effects on EP.

## 4. Conclusions

A novel phosphorus-nitrogen-containing ionic liquid was designed and the ionic liquid-modified NH_2_-MIL-101(Al) composite was prepared through a facile method. This proves that the obtained IL@NH_2_-MIL-101(Al) is able to greatly improve the fire safety of EP at a low addition of 3.0 wt. %. The LOI value of EP/IL@NH_2_-MIL-101(Al) reached 29.8%. The emissions of heat, smoke and CO were obviously decreased after the addition of IL@NH_2_-MIL-101(Al) to EP matrix. In comparison to the results for neat EP and the EP/NH_2_-MIL-101(Al) composite, the enhanced flame retardant performance of EP/IL@NH_2_-MIL-101(Al) was attributed to the synergistic role of IL and NH_2_-MIL-101(Al), which play a crucial role in gas phase and condense phase. On the one hand, the phosphorus-nitrogen-containing ionic liquid is able to trap the radicals and promote the char forming process in order to suppress EP combustion. On the other hand, the NH_2_-MIL-101(Al) framework acts as a barrier, restraining the transmission efficiency of heat and combustible material. Combing the unique properties of IL and NH_2_-MIL-101(Al), the IL@NH_2_-MIL-101(Al) composite can be implemented as an effective flame retardant for EP. In addition, this strategy provides a practical method for developing other advanced MOF composites for flame retardant application.

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
