# Peer review of "Novel Phosphorus-Nitrogen-Containing Ionic Liquid Modified Metal-Organic Framework as an Effective Flame Retardant for Epoxy Resin"

_polymers, 2020, doi:10.3390/polym12010108_

Round 1
Reviewer 1 Report
The work of “Novel Phosphorus-Nitrogen Containing Ionic Liquid 3 Modified Metal-Organic Framework as an Effective 4 Flame Retardant for Epoxy Resin” done by Huang et al. is very interesting and perfect and definitely will be fruitful for the readership of polymers. I strongly recommend the publication of this work after addressing the minor issues below.
Proof-read the paper after complete revision and consider the grammatical points and typos. Please, compare your work with the recent literature articles in the Result and Discussion section
Author Response
Comments to the Author
The work of “Novel Phosphorus-Nitrogen Containing Ionic Liquid Modified Metal-Organic Framework as an Effective Flame Retardant for Epoxy Resin” done by Huang et al. is very interesting and perfect and definitely will be fruitful for the readership of polymers. I strongly recommend the publication of this work after addressing the minor issues below.
Proof-read the paper after complete revision and consider the grammatical points and typos. Please, compare your work with the recent literature articles in the Result and Discussion section.
Answer:
Thanks for the comment.
(1) The grammatical points and typos have been revised in the revised manuscripts.
(2) The comparison with the concerned phosphorus-nitrogen based flame retardants in recent literature articles has been added in the Result and Discussion section in the revised manuscripts (Table 4, Lines 304-310, Page 11). The concerned flame retardants in literatures were summarized in Table 4. It can be seen that the IL@NH2-MIL-101(Al) exhibits excellent flame retardant property on EP at low addition. As far as we know, this is the first report of IL modified MOF composite as effective flame retardant. Thus, modification method offers a new way to prepare other advanced flame retardants.
Reviewer 2 Report
The paper “Novel Phosphorus-Nitrogen Containing Ionic Liquid Modified Metal-Organic Framework as an Effective Flame Retardant for Epoxy Resin” by R. Huang et al.
The paper reports the synthesis of the novel composite based on NH2-MIL-101(Al)metal-organic framework and a phosphorus nitrogen containing ionic liquid ([DPP-NC3bim][PMO]) as flame retardant for widely used epoxy resin. The goals of this paper are practically relevant, and the results are interesting.
However, before publication the following issues should be adressed. My recommendation is major revision. Some additional experiments should be carried out.
The caption for the Scheme 1 should be checked.
2 P. 2, lines 67-68. The phrase “On the other hand, the organic ligand of NH2-MIL-101(Al) is helpful for increasing the compatibility with EP, and provides 68 flame retardant elements.” Should be clarified. P. 3, line 118. The same for the phrase “Then different amount of prepared flame retardants were added into EP.”
The IR spectrum of the studied ionic liquid should be presented on the Figure 2. The paper lacks the results of the elemental analysis of the ionic liquids, its composite with MIL, EP/MIL, and IL@EP/MIL materials. EA needs for the confirmation of their composition. The IL content in MIL and The NMR analysis should be done for the identification of the synthesized ionic liquid. The results on N2-low temperature adsorption for the synthesized NH2-MIL-101(Al) and composite on its basis are missed in the paper. The location of IL in the NH2-MIL-101(Al) matrix should discussed taking into account the physico-chemical characterization. Possibly, TEM characterization of the synthesized composites based on this MOF should be helpful.
Round 2
Reviewer 2 Report
The authors took into account all comments.
Thus, the paper can be published in the present form.